# Humidity Sensor Based on a Long-Period Fiber Grating Coated with Polymer Composite Film

**DOI:** 10.3390/s19102263

**Published:** 2019-05-16

**Authors:** Yunlong Wang, Yunqi Liu, Fang Zou, Chen Jiang, Chengbo Mou, Tingyun Wang

**Affiliations:** 1Key Laboratory of Specialty Fiber Optics and Optical Access Networks, Joint International Research Laboratory of Specialty Fiber Optics and Advanced Communication, Shanghai Institute for Advanced Communication and Data Science, Shanghai University, Shanghai 200444, China; lightsmile@shu.edu.cn (Y.W.); ZF-198166@sohu.com (F.Z.); jiangchen17@shu.edu.cn (C.J.); mouc1@shu.edu.cn (C.M.); tywang@shu.edu.cn (T.W.); 2Department of Information Engineering, Hebei GEO University, Shijiazhuang 050031, China

**Keywords:** long-period fiber grating (LPFG), polymer coating, humidity sensor, PEG, PVA

## Abstract

We demonstrate a simple and highly sensitive optical fiber relative humidity (RH) sensor based on a long-period fiber grating (LPFG) coated with polyethylene glycol (PEG)/polyvinyl alcohol (PVA) composite films. The resonance wavelength of the LPFG is sensitive to environmental humidity due to the change in effective refractive index caused by the strong surface absorption and desorption of the porous PEG/PVA coatings. The sensor is sensitive in a wide range from 50% to 95% RH, with a highest sensitivity of 2.485 nm/%RH in the range 50–75% RH. The proposed RH sensor has the advantages of compact size, good reversibility, and stability, which makes it attractive for high-humidity environments.

## 1. Introduction

Relative humidity (RH) is an important magnitude in many fields, such as agricultural production, food storage, pharmaceuticals, health monitoring, and meteorological services. Traditional RH sensors are electric humidity sensors which consist of two main sensing parts: Electric capacity and electric resistance. They usually show hysteresis when placed in a high-humidity environment for a long time and could be affected by electromagnetic interferences. Therefore, much attention has been paid on optical fiber humidity sensors [1,2] due to their advantages such as ease of fabrication, light weight, remote measurement, high sensitivity, and most importantly, electromagnetic immunity.

Various types of optical fiber RH sensors have been proposed with different approaches. Firstly, the amplitude-based techniques detect the change of the optical intensity [3,4,5]. Secondly, the wavelength shift technique is immune to power fluctuation noise, when compared with the optical intensity technique [6,7,8,9,10,11,12,13]. The other techniques [14,15,16] are not used very often and are different from the usual methods, such as detecting phase shift of the light [14] or measuring the strain on the fiber [15]. As for wavelength-based techniques, the transmission spectrum can be used to monitor the RH change of the environment. Several structures, such as cladding-etched optical fiber [6], side-polished fiber [7], fiber Bragg grating [8,9,10], Fabry–Perot interference [11], and long-period fiber grating (LPFG) [12,13] have been proposed. These structures usually have a transmission dip at a specific wavelength, which could be used to measure the surrounding refractive index (SRI).

As an optical fiber sensor, LPFG has been widely used in many areas, for the measurements of SRI [17] and temperature [18]. Zhao et al. [17] have fabricated the LPFG in thinned cladding fiber with a high index sensitivity of 7366.6 nm/refractive index unit (RIU). Zou et al. [18] have presented a new method to improve the temperature sensitivity of the LPFG. They used atomic layer deposition technology to deposit Al_2_O_3_ nanofilm on the LPFG and packaged it with the polymer material. A temperature sensitivity of up to 0.77 nm/°C was achieved. LPFG is usually insensitive to surrounding RH. Different coating methods have been studied to combine LPFG with RH sensitive films as an RH sensor. Alwis et al. [12] have presented a self-interfering LPFG coated with polyvinyl alcohol (PVA) as an RH sensor. The grating-based sensor probes showed an optimum sensing range of 40–85% RH with negligible hysteresis. Urrutia et al. [13] have fabricated a semi-coated LPFG with poly allylamine hydrochloride (PAH) and poly acrylic acid (PAA) by a layer-by-layer nano-assembly technique. The RH sensitivity is 63.23 pm/%RH in the range 20–80% RH.

There are different techniques for the fabrication of the LPFGs. The UV light writing technique is the common one adopted to fabricate LPFGs in photosensitive fibers by lots of groups [12]. The CO_2_ laser writing technique is a flexible technique for the fabrication of LPFGs thanks to its advantages such as low cost, high grating temperature stability, etc. The method can be applied to practically any fibers, and the writing process can be computer programmed to generate complicated grating profiles without using any masks [19]. The mechanism of the CO_2_-laser-written LPFGs can be attributed to the residual stress relaxation and glass structure changes due to laser heating. The LPFGs written in a commercial boron-doped fiber can give a clean transmission spectrum, just like UV-written LPFGs [19].

In this paper, we demonstrate a highly sensitive humidity sensor based on a CO_2_-laser-written LPFG coated with polyethylene glycol (PEG)/PVA composite films. The films are coated on the LPFG by spontaneous evaporation of the PEG/PVA water solution. The coated films improve the interaction between the LPFG and the external humidity environment based on the adsorption and desorption of the PEG/PVA thin films. The coated LPFG humidity sensor shows a high RH sensitivity in a wide range from 50% to 95% RH. The experimental results show good repeatability, reversibility, and stability. The sensor structure has the advantages of small size and simple fabrication, and the thickness of the film can be controlled by changing the concentration of the solution. The PEG and PVA materials have good hydrophilicity and high molecular stability. The polymer composite film has a refractive index close to that of fiber cladding, which makes the sensor highly sensitive to RH changes. Therefore, it is possible to accurately measure the external RH based on the wavelength shift of the LPFG.

## 2. Principle and Sensor Fabrication

### 2.1. Principle and Experimental Setup

According to the phase-matching condition, the relationship between the fundamental core mode and the forward propagated cladding modes of LPFG can be expressed as follows [20]:
$$\lambda_{m}=(n_{\mathrm{co}}^{\mathrm{eff}}-n_{\mathrm{cl},m}^{\mathrm{eff}})\Lambda (m=1,2,\ldots ),$$
where $\lambda_{m}$ is the resonance wavelength and Λ is the grating period. The principle of the RH sensing is based on the humidity-induced refractive index changes of the coating materials. PEG/PVA composite films were adopted to coat the LPFG. When the ambient humidity changes, the coating material swells or deswells, and the effective refractive index of the coating material changes accordingly, which leads to the resonance wavelength shift of the LPFG. Therefore, the change of RH can be characterized by the resonance wavelength shift of the coated LPFG.

The schematic experimental setup for measuring RH is shown in Figure 1a, which consists of an optical spectrum analyzer (OSA: Yokogawa AQ6370C), a temperature and humidity test chamber, and a broadband optical light source (Amonics, ASLD-CWDM-5-B-FA). The sensor probe was placed in the temperature and humidity test chamber, the transmission spectra of the sensor head was measured by the OSA when the temperature and RH were changed. The inset shows the microscope image of the composite PEG/PVA films. 

The LPFG was written in boron–germanium co-doped single-mode fiber (SMF, Fibercore PS 1250/1500) by CO_2_ laser (CO_2_-H10, Han’s Laser, ShenZhen, China) with a period of 300 μm and a length of 1.8 cm. The CO_2_ laser pulse had a frequency of 5 kHz and a wavelength of 10.6 μm. The spot beam diameter of the CO_2_ laser focused on the fiber was approximately 30 μm. The output power of the laser was around 0.6 W. The LPFG had a resonance wavelength of ~1560 nm. The mode patterns of the LPFGs were observed using a Charge-coupled Device (CCD) camera (InGaAs camera, Model C10633-23 from Hamamatsu Photonics, Hamamatsu City, Japan) [19,21]. The resonance dip was identified to be in the LP_0,8_ cladding mode. The LPFG was fixed in the glass U-groove (50 mm × Φ2 mm × 1 mm) with the heat-curing adhesive. The U-groove was fixed on the slide with the heat-curing adhesive, as shown in Figure 1b. 

### 2.2. Sensor Fabrication

PEG (Chemical Abstracts Service (CAS): 25322-68-3) and PVA (CAS: 9002-89-5) solid powder reagents were used in the experiments. The organic solvents alcohol was of analytical grade, and deionized water was used throughout the experiments. Figure 2a shows the molecular formula of the two materials. PVA molecules are easily soluble in water and have many OH groups which bond the carbon atoms with each other in the main chain to form a macromolecular scaffold [16]. The PVA film has a good uniformity and high swelling ratio. When the PVA films absorb water, its refractive index changes accordingly. Therefore, the PVA film is a good candidate as a wet sensitive material. The 3% (wt./wt.) PVA solution was prepared by mixing it with deionized water. To make sure the PVA molecules were totally dissolved, the solution was placed in a magnetic stirrer at constant temperature (95 °C) for 2 h to let the PVA molecules to fully swell.

In our experiment, relative humidity sensing is based on the wavelength shift measurement of the coated LPFG. Therefore, higher resolution can be achieved when the refractive index of the coating materials is near the refractive index of the fiber cladding. The refractive index of hydrophilic materials is generally higher than that of fiber cladding, it is very unusual for the material to have both higher hydrophilicity and lower refractive index. PEG is a polar material and was adopted to reduce the refractive index of the coating PVA films. The refractive index of PEG can be close to the index of the fiber cladding when the external humidity rises to a certain degree. The refractive index of the coating materials decreases due to absorption of water molecules. During the process of humidity change, the film swells and shrinks. As described in [22], as the RH level rises, the hydrophilic PVA film absorbs more water, thus causing the swelling of the PVA coating and a reduction in density, which finally results in a reduction in the refractive index of the coating film, and its refractive index decreases from 1.49 to 1.35 when the RH increases from 20% to 95% RH [23]. In our experiments, we use the PEG/PVA composite films as the coating materials—PEG and PVA have similar water absorption characteristics. When the humidity changes, the film absorbs and desorbs water molecules due to the interaction of the film and water molecules, which changes the overall refractive index [23,24]. When the humidity changes in a certain range, the refractive index of the films falls below the refractive index of the fiber cladding. Therefore, the resonance wavelength of the LPFG shifts with the surrounding humidity. The PEG decreases the effective index of the composite films so that the wavelength shift measurement of the LPFG can be adopted. Compared with the sensors coated with PVA film only, the wavelength shift measurement can avoid the interference caused by fluctuation of the light source and noise [22,24]. 

According to the theoretical and experimental results, changes in the thickness of the nanofilms coated on the LPFGs could change the refractive index sensitivity of the LPFGs dramatically when the thickness of the nanofilms is a few hundreds of nanometers (for example, 100~300 nm) [21,25]. In our experiments, the LPFGs were coated by a film with a thickness of ~900 nm. For film thickness of 900–1000 nm, small changes in the film thickness have little effect on the sensing characteristics of the coated LPFGs [21,25].

Both PVA and PEG materials are water-soluble polymers and have many OH and O groups bonded in the backbone chain. The 15% (wt./wt.) PEG solution was prepared by dissolving the PEG powder in deionized water. The solution was prepared at room temperature and stirred until it became clear and transparent. Then, the PVA and PEG solutions were mixed together. To obtain a uniform dispersive solution, the PEG/PVA solution required an hour of ultrasonic processing. The PEG/PVA films were coated on the LPFG by evaporation of the water molecules of the PEG/PVA solution in a dry oven. Before the coating process, the LPFG sample in the glass U-groove was cleaned repeatedly with deionized water and alcohol and dried with a vacuum oven. We used a high-precision pipette to inject the mixed solution into the glass U-groove, ensuring that the same volume of mixed solution was injected every time during the process. After the mixture solution was dropped into the glass U-groove, the coated LPFG was placed in the drying oven for about 12 h to ensure enough water evaporation. The PEG/PVA films were then coated on the LPFG surface. Since the glass U-groove filled with the PEG/PVA mixed solution was horizontally placed in the drying oven and the solute was uniformly dispersed in the solution, the evaporation of water molecules was a slowly and evenly changing process. Therefore, we believe that the film was uniformly coated on the surface of the LPFG after the water molecules were completely evaporated. The thickness of the polymer film was ultimately determined by the concentration of the mixed solution [22]. Figure 2b shows the schematic configuration of the RH sensor coated with the PEG/PVA composite film. The main factor determining the film thickness was the concentration of the material when the solution was prepared because the volume of the glass U-groove was fixed, so, the greater the concentration, the greater the film thickness. The Raman spectrum of the PEG/PVA films coated on the LPFG was excited by a 598 nm laser (LabRAM HR Evolution, HORIBA Jobin Yvon, Paris, France) and measured at room temperature. From the measured spectrum, we can clearly see the absorption peaks of polar groups in the polymer material. They can interact with water molecules during the change of humidity, thus changing the refractive index of the material. 

The scanning electron microscopy (SEM) image of the coated LPFG is shown in Figure 3. The surface topographies can be seen from Figure 3a,b, and the first one is coated with pure PVA with a smooth surface, while the second one is coated with PEG/PVA composite films, which is highly corrugated and wrinkled. It can be believed that the surface wrinkles and holes will facilitate the entry of water molecules [22,26] and it will be more conducive to humidity sensing. Figure 3c is the cross-section of the coated grating. By blending PEG/PVA films with different concentration schemes, polymer films with different thicknesses were coated on the surface of the LPFGs. The sensitivity of humidity measurements may be different for the coated LPFGs with different film thickness. We have done experiments with the sensing LPFGs with different film thicknesses and the experimental results suggested that the film thickness may have an influence on the performance of the sensor. Figure 3d shows the SEM picture of the sensor with the best humidity sensitivity, whose film thickness was measured to be ~906.3 nm. The coating method demonstrates very good repeatability. The non-uniformity across the coating structure has little effect on the sensitivity of the coated LPFGs.

## 3. Experimental Results and Discussion

We measured the refractive index sensitivity of the LPFG before it was coated with polymer, as shown in Figure 4. The principle of the humidity measurement is based on the humidity-induced refractive index change of the coating materials of the LPFGs. When the surrounding humidity is changed, the refractive index of the coating materials is changed accordingly, thus the resonance wavelength of the LPFG changes due to the surrounding refractive index changes. It can be seen from Figure 4b that the dependence of the resonance wavelength of the LPFG on the surrounding refractive index is nonlinear. The LPFG shows higher refractive index sensitivity when the surrounding refractive index is close to the refractive index of the fiber cladding. In the experiment, the measured trend of the humidity sensing was in good agreement with the refractive index sensing of the LPFG.

During the experiment, the temperature of the RH chamber was fixed at 30 °C. The RH was changed from 50% to 95% with a step of 5% RH. When the humidity value is less than 50%, the number of water molecules absorbed by the materials is not large enough to decrease the refractive index of the coating materials to fall below the refractive index of the grating cladding, so the resonance dip of the LPFG will not shift with the change of the humidity. By finding coating materials with a lower refractive index, the dynamic range of the humidity measurements could be increased. The change of humidity was observed by measuring the wavelength shift of the resonance dip when the humidity reached a stable value at the set value. Figure 5a shows the experimental results of the RH measurements of the LPFG coated with pure PEG films. A comparison of the LPFGs coated with pure PVA films is also shown in Figure 5a. It can be observed that the pure PVA was almost insensitive to humidity while the pure PEG was very sensitive to humidity with a nonlinear response. PEG/PVA composite films were adopted to improve the sensitivity and linearity of the RH measurements.

Figure 5b shows the humidity sensing characteristics of the LPFGs coated by the composite films with different PEG proportions in the humidity range from 50% to 95% RH. It can be seen that the composite film has a larger refractive index when the proportion of the PEG is less in the films. When the humidity is changed at a lower RH value, the refractive index of the film is higher than the fiber cladding and the light of the LPFG could be coupled from the core mode to the radiation modes [27,28] for the LPFGs coated by the composite film with a small proportion of PEG. The break point of the measured results for the LPFGs coated by the films with a PEG proportion of 1/3 and 1/2 is because the light is coupled into radiation modes instead of cladding modes due to the higher refractive index of the coating films at lower RH value. In the experiments, the film with a PEG proportion of 2/3 was used as the coating material because the coated LPFGs showed a high RH sensitivity and a better linearity in the high RH range.

The refractive index change of the coating polymer could be estimated by comparing the wavelength shifts of the bare LPFG (Figure 4) and coated LPFG (Figure 5). For example, when the resonance wavelength of the coated LPFG is ~1540 nm (95% RH), it can be seen from Figure 4b that the refractive index of the coating polymer is about 1.375. When the resonance wavelength of the coated LPFG is ~1488 nm (60% RH), it can be seen from Figure 4b that the refractive index of the polymer is about 1.457. Therefore, the refractive index of the coating polymers is estimated to change from ~1.375 to ~1.457 for the humidity changes from 95% to 60%.

Figure 6 shows the transmission spectra of the coated LPFGs when the RH is ascending and descending with a PEG proportion of 2/3. For the conventional LPFGs, the grating contrast decreases when the SRI is close to that of the fiber cladding. In order to measure the wavelength shift of the LPFG accurately, over-coupled LPFGs can be fabricated, which were reported to have higher grating contrast with the SRI close to that of the fiber cladding [21]. In the humidity range of 50–55% RH, the corresponding transmission dip position does not appear obvious due to the lower grating contrast with SRI close to that of the fiber cladding. The LPFG RH sensor works well in the 50–95% RH range. To increase the measurement accuracy in the range from 50% to 55% RH, over-coupled LPFGs could be adopted. For the LPFGs coated with polymers, with the increase of the relative humidity, the refractive index of the polymer becomes lower than that of the fiber cladding, the LPFG spectra becomes deeper and narrower, which is consistent with the theoretical analysis and reported experimental results [28,29]. When the external environment RH increases, the water molecules in the air will increase. At this time, the balance of the water molecules absorbed by the original polymer is broken and the water absorption process will reoccur. More water molecules will decrease the refractive index of the polymer, which makes the resonance wavelength of the coated LPFG red-shift. Similarly, the resonance wavelength blue-shifts when the RH decreases, because the films desorbs the water molecules, which makes the effective refractive index of the coating materials increase.

Figure 7a shows the fitting curve of the resonance wavelength on the RH under the ascending and descending humidity process. We also measured the RH sensitivity in the range from 50% to 75% RH. The refractive index of the coating materials was close to the effective refractive index of the cladding modes in the above humidity range, the humidity sensitivity was very high. Additionally, the measured sensitivity was 2.485 nm/%RH in the range from 50% to 75% RH. To prove that the proposed PEG/PVA composite films can enhance the RH sensitivity, the RH sensitivity of the bare LPFG was also measured for comparison. The uncoated LPFG had a very low RH sensitivity of 0.00182 nm/%RH, which can be considered negligible. The coated LPFG showed a very high sensitivity of 2.485 nm/%RH, which is about 1365 times higher than that of the bare LPFG. We can estimate the refractive index changes of the coating materials by comparing the experimental results shown in Figure 4b and Figure 7a. Therefore, the experimental results show that the refractive index of the film changes correspondingly during the change of relative humidity.

To investigate the temperature cross-sensitivity of the proposed humidity sensor, the temperature sensitivity of the sensors was measured when the sensors were set at different temperatures from 20 to 50 °C with a step of 5 °C. The RH was fixed at a specific value when the temperature was changed. For the humidity of 66% and 70% RH, the bare LPFG showed a temperature sensitivity of −0.43 nm/°C. When the coated LPFGs were set under different relative humidity environments, the temperature sensitivity of the sensor was much lower than that of the bare LPFGs. Moreover, when the humidity was in the range of 66–70% RH, the temperature cross-sensitivity was almost negligible, as shown in Figure 7b. The LPFG written in the boron–germanium co-doped fiber had negative temperature sensitivity [30] and the PEG/PVA coating film had the opposite thermo-optic coefficient, which compensated with each other when the environment temperature changed for the coated LPFGs. The experimental results show that temperature fluctuation had a low effect on the humidity sensing. 

We coated the LPFGs with films consisting of different compositions of PEG/PVA. The experimental results of the humidity sensing measurements suggested that the LPFGs coated with the composite polymers with different combination ratios have different humidity sensitivities.

Figure 8a shows the linear fitting results of two different measurements for the same sensor sample. The test time interval was approximately one month between the first and the second tests. It was obvious that the two curves were highly coincided. The difference in the RH sensitivity was small, only 0.00231 nm/%RH. Therefore, the proposed sensor had high repeatability. At the same time, the stability of the fabricated RH sensor was measured and is depicted in Figure 8b. The resonance wavelength remained almost unchanged for three RH values within one hour. Therefore, the proposed sensor demonstrated perfect repeatability and stability.

A comparison of wavelength-based fiber optic humidity sensors coated with different types of films is shown in Table 1. The table shows various wavelength-based fiber optic RH sensors, most of these structures were formed through optical interference like Fabry–Perot interference and Mach–Zehnder interference. Sensors with different structures usually have a transmission spectrum at a specific wavelength and can measure the RH changes in a dynamic range. For example, Nafion film temperature/humidity sensing based on optical fiber Fabry–Perot interference shows a relative humidity sensitivity of 3.78 nm/%RH for the range of 30–85% RH [11], but, the system structure of the proposed sensor is difficult to fabricate. The sensitivity of the PEG/PVA-coated LPFG is higher than that for most of the other configurations like Mach–Zehnder interferometer, side-polished fiber, U-shaped microfiber interferometer, polymer microfiber rings, etc. Even though its sensitivity is lower than that of the Fabry–Perot interference-based fiber optic sensor, the proposed sensor has the advantage of easy fabrication. The coated LPFG sensor shows much lower temperature cross-sensitivity than the bare LPFG, which has little effect on RH sensing, as shown in Figure 7b. We believe the LPFG coated by PEG/PVA film could be an ideal candidate for high-performance fiber RH sensing.

The resolution of the humidity measurements is dependent on the wavelength resolution of the wavelength shift measurements. In our experiment, we used an OSA with a wavelength resolution of 0.02 nm to measure the wavelength shift of LPFGs. The resolutions of the sensors could be improved by using equipment with higher-wavelength resolution. Since the temperature and humidity test chamber (ESL-04KA, ESPEC, GuangZhou, China) used in our experiment was quite large, it took about 5 min for the instrument itself to rise from one set RH value to another one. We did not have equipment to measure the response time of the RH sensor. The response time of the sensor is related to the film thickness and the coating materials. Our sensor has almost the same film thickness as the one reported in the reference [22]. The sensor coated by the PVA film was reported to have a fast response time of 630 ms. Therefore, we could estimate that our sensor based on the LPFGs coated with PEG/PVA composite film will also have a fast response time.

## 4. Conclusions

In summary, a high-sensitivity optical fiber humidity sensor based on PEG/PVA film-coated LPFG was fabricated. Experimental results show that the sensor with a coating thickness of 906.3 μm has a high humidity sensitivity in a wide range of 50–95% RH. The measured results suggest that the proposed sensor has lower temperature cross-sensitivity. The sensor exhibits good linearity, high stability, and reversibility in the wide working range, which makes it a better choice for RH sensing measurement.

## Figures and Tables

**Figure 1 sensors-19-02263-f001:**
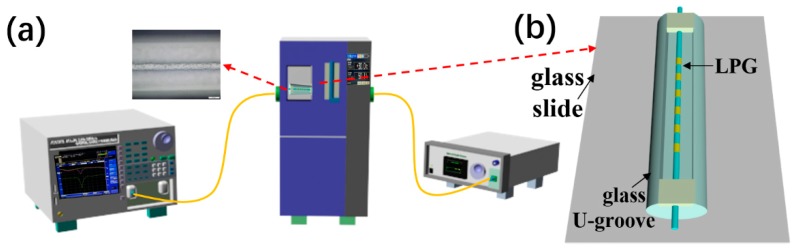
(**a**) Experimental setup to monitor the response of the relative humidity (RH) sensor. The inset shows the microstructure of the polyethylene glycol (PEG)/polyvinyl alcohol (PVA) composite film; (**b**) the packaged structure used in the deposition of the PEG/PVA film.

**Figure 2 sensors-19-02263-f002:**
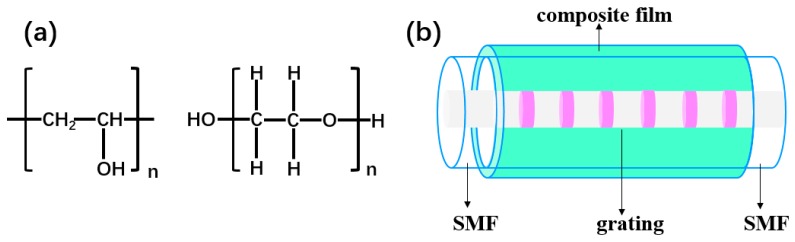
(**a**) Chemical structures of the composite materials. Polyvinyl alcohol is on the left, polyethylene glycol is on the right; (**b**) schematic configuration of the long-period fiber grating (LPFG) RH sensor coated with a PEG/PVA composite film. SMF—single-mode fiber.

**Figure 3 sensors-19-02263-f003:**
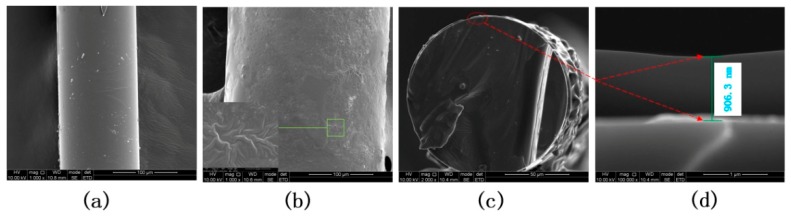
SEM images of the coated LPFG. (**a**) Fiber coated with pure PVA film; (**b**) fiber coated with PEG/PVA composite film; (**c**) cross-section of the coated LPFG; (**d**) film thickness of the coated film.

**Figure 4 sensors-19-02263-f004:**
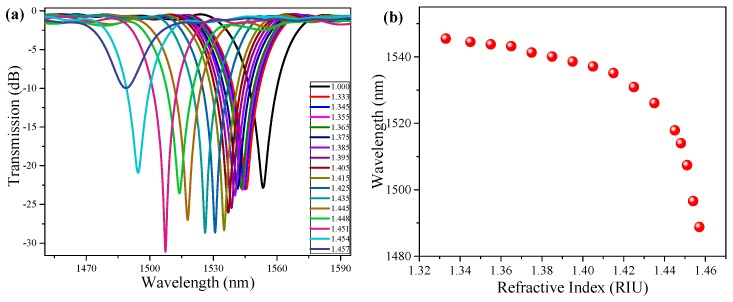
Refractive index sensing characteristics of LPFG: (**a**) Transmission spectra of the LPFG with different surrounding refractive index; (**b**) dependence of the resonance wavelength on the surrounding refractive index.

**Figure 5 sensors-19-02263-f005:**
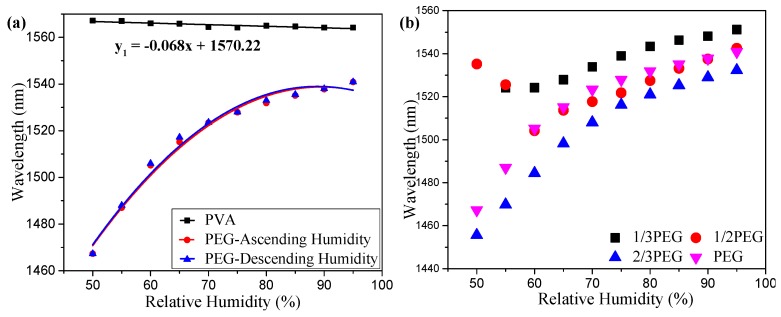
(**a**) A comparison of different materials for relative humidity sensing. (**b**) Humidity sensing characteristics of the LPFGs coated by the films with different PEG proportions in the humidity range from 50% to 95% RH.

**Figure 6 sensors-19-02263-f006:**
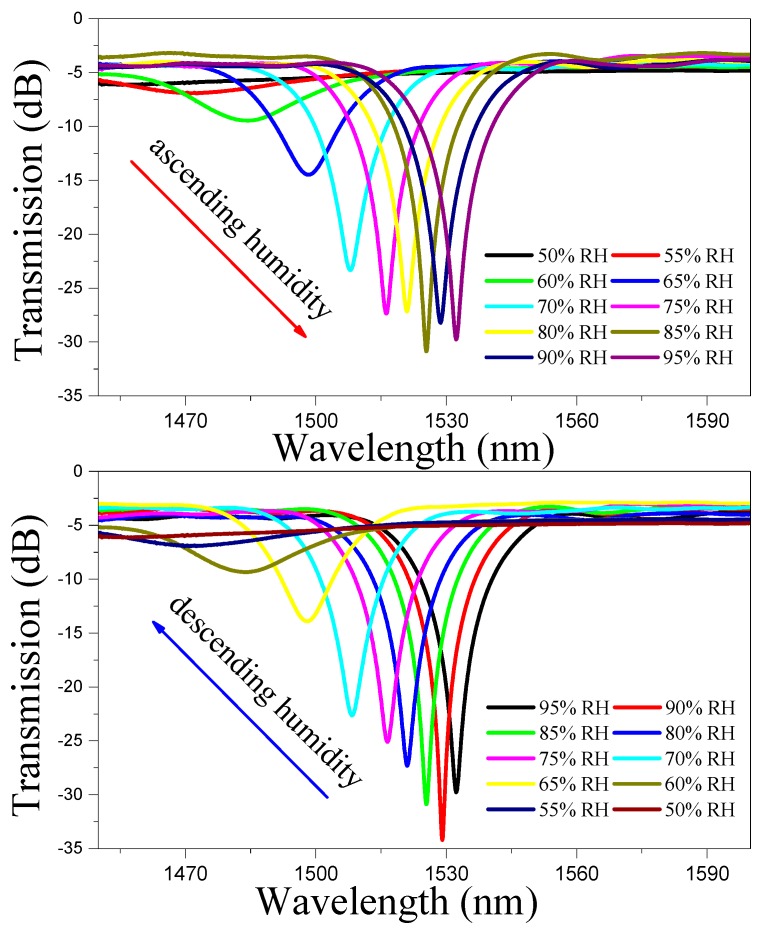
The transmission spectra of the coated LPFGs with a PEG proportion of 2/3 when the relative humidity is ascending and descending.

**Figure 7 sensors-19-02263-f007:**
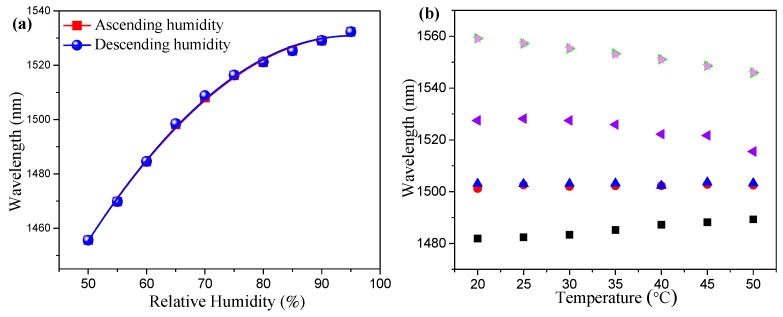
(**a**) The fitting line of the resonance dips under the ascending and descending humidity process. (**b**) Temperature response of the LPFGs without and with polymer coatings at different humidity. (►) 66% RH without PEG/PVA composite film coating, (★) 70% RH without PEG/PVA composite film coating, (■) 60% RH with PEG/PVA composite film coating, (●) 66% RH with PEG/PVA composite film coating, (▲) 70% RH with PEG/PVA composite film coating, (◄) 90% RH with PEG/PVA composite film coating.

**Figure 8 sensors-19-02263-f008:**
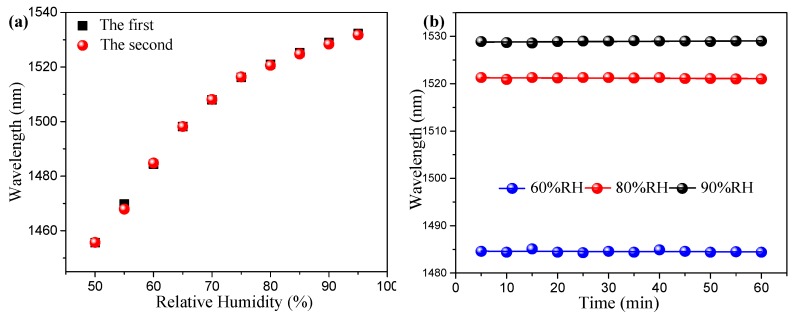
(**a**) Repeatability verification of the first and second experiments. (**b**) The stability test of the proposed PEG/PVA-coated RH sensor.

**Table 1 sensors-19-02263-t001:** Comparison between wavelength-based fiber optic humidity sensors.

Reference	Configuration	Dynamic Range (%RH)	Wavelength Shift (nm)	Sensitivity (nm/%RH)
This paper	Polymer-coated LPFG	25 (50–75)	62.125	2.485
[6]	Cladding-etched optical fiber	70 (20–90)	133	1.9
[7]	Side-polished fiber	12.6 (85–97.6)	11.529	0.915
[11]	Fabry–Perot interference	55 (30–85)	207.9	3.78
[31]	Polymer optical fiber	80 (10–90)	0.5848	0.00731
[32]	U-shaped microfiber interferometer	60 (30–90)	6.882	0.1147
[33]	Mach–Zehnder interferometer	80 (10–90)	9.568	0.1196
[34]	Polyimide film-coated optical fiber	40 (40–80)	52.36	1.309
[35]	Polymer microfiber rings	66 (5–71)	32.43	0.49
[36]	singlemode-multimode-singlemode fiber structure	50 (35–85)	11.15	0.223
[37]	Thin film-based one-dimensional photonic crystal	73 (11–84)	21.9	0.3
[38]	SiO_2_ nanoparticle-coated S-taper fiber	11.4 (83.8–95.2)	13.36	1.1718

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
