# Peer review of "Humidity Sensor Based on a Long-Period Fiber Grating Coated with Polymer Composite Film"

_sensors, 2019, doi:10.3390/s19102263_

Round 1

Reviewer 1 Report

1) the authors say that the process of the evaporating the films on the fiber was natural, what does it mean and how was the uniformity of the film thickness ensures during this process? Is there any impact of the film thickness, or its non-uniformity across the coating structure on the performance of the sensor?

2) The wavelength peaks for the humidity range of 50% RH  and 55% RHin figure 6 are nominal. The authors have not considered them for characterizing the temperature response of the sensor in the figure 7b. The sensor seems to work well in a short range of 60-95%RH only.

3) The graph legend in figure 7b must be reconstructed.

4) in line 112 compare, should be read compared.

5) paraphrase line 260

Author Response

Thanks very much for your and the reviewers’ kindly help. We have revised the manuscript according to the reviewers’ comments carefully. The changes we have made are marked with underline in the revised manuscript. We wish the revised version could meet the requirement of publication in Sensors. Look forward to hearing from you again.

Reviewer 2 Report

This paper describes the fabrication and development of a long period fibre grating (LPFG) sensor coated with polymers that aims to detect relative humidity. Although the technical aspects of the work are sound the novelty is questionable as many other LPFG based sensors modified with the same or similar materials have been reported. Authors need to clearly state what is the novelty in this work. It seems that the main difference is in the fabrication of LPFG using CO2 laser, but this is not discussed in sufficient details. It is also not very clear why the refractive index (RI) of the coating would decrease with the humidity increase. The explanation of the sensing mechanism needs to be strengthened. Please see detailed comments below that need to be addressed before I can recommend this paper for publication in Sensors:

1.   The introduction needs a discussion on different types of LPFG. In this work, you are using CO2 to fabricate LPFG but the most common method is to use UV light, discussion on the advantages and disadvantages of these methods need to be mentioned.

2.   In equation 1 wavelength is also discreet parameter and will change with the change of m, so should be lm.

3.   The description of the mechanism is not very accurate. If there is a swelling and shrinking, why the RI should change?  LPFG is sensitive to the film thickness as well.

4.   There are more papers on LPFG for humidity sensing. For example: (Measurement 46 (2013) 4052–4074);

5.   What are the parameters of CO2 laser (power, beam size, etc.) used for LPFG fabrication?

6.   Why for Peg material RI with the absorption of water would decrease?

7.   Why in Figure 5b for 1/2PEG with the increases of RH wavelength decreases (i.e. RI increase) and then wavelength increases (i.e. RI decreases).

8.   LPG is sensitive also to the thickness increases, you can’t use wavelength shift to measure the RI change directly without knowing what thickens the film has.

9.   Figure 6, which polymers were used to coat the LPFG?

10. “When the external environment RH increases, the composite films absorbed the water molecules, the effective refractive index of the coating materials will decrease, which makes the resonance wavelength of the coated LPFG red-shift” why RI would decrease with the increase of the RH?

11. Important parameter if any sensor is response time, why this parameter is not indicated? What is the response time?

Author Response

(The authors gave the same response as above.)

Round 2

Reviewer 2 Report

all comments have been addressed.